# Single-phase perovskite oxide with super-exchange induced atomic-scale synergistic active centers enables ultrafast hydrogen evolution

Jie Dai[1,8], Yinlong Zhu[2,8✉], Hassan A. Tahini[3], Qian Lin[2], Yu Chen[4], Daqin Guan[1], Chuan Zhou[1], Zhiwei Hu[5], Hong-Ji Lin[6], Ting-Shan Chan[6], Chien-Te Chen[6], Sean C. Smith[3], Huanting Wang[2], Wei Zhou[1] & Zongping Shao[1,7✉]

The state-of-the-art active HER catalysts in acid media (e.g., Pt) generally lose considerable catalytic performance in alkaline media mainly due to the additional water dissociation step. To address this issue, synergistic hybrid catalysts are always designed by coupling them with metal (hydro)oxides. However, such hybrid systems usually suffer from long reaction path, high cost and complex preparation methods. Here, we discover a single-phase HER catalyst, $SrTi_{0.7}Ru_{0.3}O_{3-\delta}$ (STRO) perovskite oxide highlighted with an unusual super-exchange effect, which exhibits excellent HER performance in alkaline media via atomic-scale synergistic active centers. With insights from first-principles calculations, the intrinsically synergistic interplays between multiple active centers in STRO are uncovered to accurately catalyze different elementary steps of alkaline HER; namely, the Ti sites facilitates nearly-barrierless water dissociation, Ru sites function favorably for OH* desorption, and non-metal oxygen sites (i.e., oxygen vacancies/lattice oxygen) promotes optimal H* adsorption and $H_2$ desorption.

[1] State Key Laboratory of Materials-Oriented Chemical Engineering, College of Chemical Engineering, Nanjing Tech University, 210009 Nanjing, P.R. China. [2] Department of Chemical Engineering, Monash University, Clayton, VIC 3800, Australia. [3] Integrated Materials Design Laboratory, Department of Applied Mathematics, Research School of Physics and Engineering, Australian National University, Canberra 2601, Australia. [4] Monash Centre for Electron Microscopy, Monash University, Clayton, VIC 3800, Australia. [5] Max Planck Institute for Chemical Physics of Solids, Nothnitzer Strasse 40, Dresden 01187, Germany. [6] National Synchrotron Radiation Research Center, 101 Hsin-Ann Road, Hsinchu 30076, Taiwan. [7] Department of Chemical Engineering, Curtin University, Perth, WA 6845, Australia. [8] These authors contributed equally: Jie Dai, Yinlong Zhu. ✉email: yinlong.zhu@monash.edu; shaozp@njtech.edu.cn

Faced with excessive fossil fuel consumption and associated environmental issues, the exploitation of sustainable and green energy sources is urgently demanded[1]. As an environmentally friendly and high-density energy carrier, hydrogen ($H_2$) has been highly regarded as an attractive fuel alternative to conventional fossil fuels[2,3]. At present, $H_2$ is mainly produced by steam methane reforming and gasification of coal, which generally involves the use of fossil fuels and high temperature, giving rise to energy inefficiencies, high costs, low-purity of hydrogen products, and greenhouse gas $CO_2$ emissions[4,5]. To overcome this dilemma, an alternative promising technology to produce high-purity $H_2$ in a green and sustainable way is via electrochemical water splitting, which employs renewable electricity as energy input[3–7]. However, the rate of the hydrogen-evolution reaction (HER) in electrocatalytic water splitting is sluggish and requires efficient electrocatalysts to expedite the HER process[6,7]. In view of the fact that most oxygen-evolution reaction (OER) electrocatalysts on the counter electrode are subjected to corrosion in acid, alkaline water electrolysis is more competitive and widely adopted in industry for scalable hydrogen generation[8,9]. Consequently, substantial efforts have been devoted toward the design of efficient HER electrocatalysts in basic media to integrate with basic OER counterpart for practical water electrolysis application. In general, alkaline HER is believed to proceed via two steps, i.e., the either Volmer–Heyrovsky or the Volmer–Tafel pathways (Volmer: $H_2O + e^- \rightarrow H^* + OH^-$; Heyrovsky: $H_2O + H^* + e^- \rightarrow H_2 + OH^-$; Tafel: $H^* + H^* \rightarrow H_2$)[4,9,10]. Compared with the acid media, the kinetics of HER in alkaline media is substantially slower due to the extra water-dissociation energy barrier in the Volmer step, rendering it more challenging[3,4,9,10]. For instance, the most active HER catalyst in an acidic environment, platinum (Pt), shows two to three orders of magnitude lower activity than that in the basic environment due to the unfavorable water-dissociation kinetics on Pt[10]. To address this issue, the synergistic hybrid catalyst was originally designed via coupling Pt with metal (hydro)oxides by the Markovic group[10]. More specifically, they electrodeposited an additional water-dissociation promoter, namely $Li^+$-intercalated $Ni(OH)_2$, into Pt to boost the initial water dissociation in alkaline media and the production of hydrogen intermediates, which are then adsorbed on the Pt surface and recombine to molecular hydrogen. Motivated by their pioneering work, various hybrid systems, e.g., Pt-Co(OH)$_2$/carbon cloth[11], Pt-CoS$_2$/carbon cloth[12], Ni$_3$N/Pt nanosheet[13], Pt$_3$Ni-NiS/carbon[14], and Pt$_3$Ni/NiO$_x$[15], have been developed as efficient HER catalysts in alkaline media owing to the synergistic effect of multiple components, as schematically illustrated in Fig. 1a.

Notwithstanding these efforts, this strategy still suffers from the problems of high cost and the complicated fabrication processes (e.g., electrodeposition, wet-chemical routes), which are difficult for cost-effective and large-scale application. In addition, the unexpected long reaction paths resulting from the random distribution of multiple active sites among these hybrid systems would give rise to undesirable transport and reaction resistance[16]. Moreover, few recent studies have shed light on the key role of the adsorption of $OH^-$ ions in the alkaline HER electrocatalysis besides the water dissociation and $H^*$ adsorption on the catalyst[3,8,17–19]. Strong adsorption of $OH^-$ on catalysts can retard the water dissociation process and poison the active sites for subsequent $H_2$ combination[17–19]. The binding energy of $OH^-$ should be optimal for a high-performance catalyst so that more active sites can be exposed for the full reaction to proceed[20]. Based on the aforementioned design paradigms, the creation of synergistically catalytic centers in short reaction paths for favorable $H_2O$ dissociation, $OH^*$ desorption and $H^*$ adsorption simultaneously within a cost-effective single-phase catalyst (as illustrated in Fig. 1b), is highly desirable to boost alkaline HER kinetics; yet still, such catalysts with intrinsically catalytic synergy have not been reported so far.

During the past decade, transition metal oxides (TMOs) have aroused considerable interest as a group of promising OER electrocatalysts under alkaline conditions[21,22]. Some TMOs were also reported to hybridize with metals to promote the water-dissociation process[15,23]. Nevertheless, pure TMOs are principally inactive toward alkaline HER because of their unsatisfactory hydrogen-adsorption energy and intrinsically low electrical conductivity[4,24,25]. Among TMOs, perovskite oxides containing more than one metal have been reported for various applications by virtue of their structural and compositional flexibility[22,26–28]. The multiple ions (including metal and oxygen ions) and variable structures of perovskite oxides can bring about some unique electronic and conductive properties, which then modulate the binding energies of reaction intermediates and electron-transport behavior, and consequently their electrocatalytic activities[16,22,26,29–31]. Therefore, designing an ideal perovskite system with multiple catalytic sites, which are tailored for targeting steps in alkaline HER electrocatalysis, may be viable, but still remains a great challenge and yet to be realized.

Here, we successfully design and prepare a single-phase perovskite oxide electrocatalyst SrTi$_{0.7}$Ru$_{0.3}$O$_{3-\delta}$ (STRO) by a facile and mature solid-phase reaction method, featured with the intrinsically atomic-level catalytic synergy for alkaline HER electrocatalysis. Remarkably, an unusual super-exchange effect in

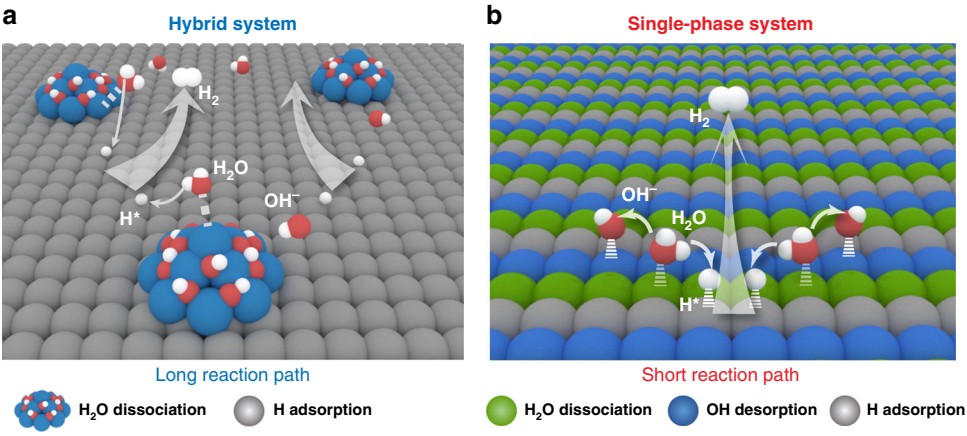

**Fig. 1 Schematic description of the reaction pathways on two-type synergistic catalysts for alkaline HER electrocatalysis. a** The conventional hybrid system by coupling noble metals with metal (hydro)oxides. **b** The conductive single-phase system with intrinsically atomic-scale synergistic active centers.

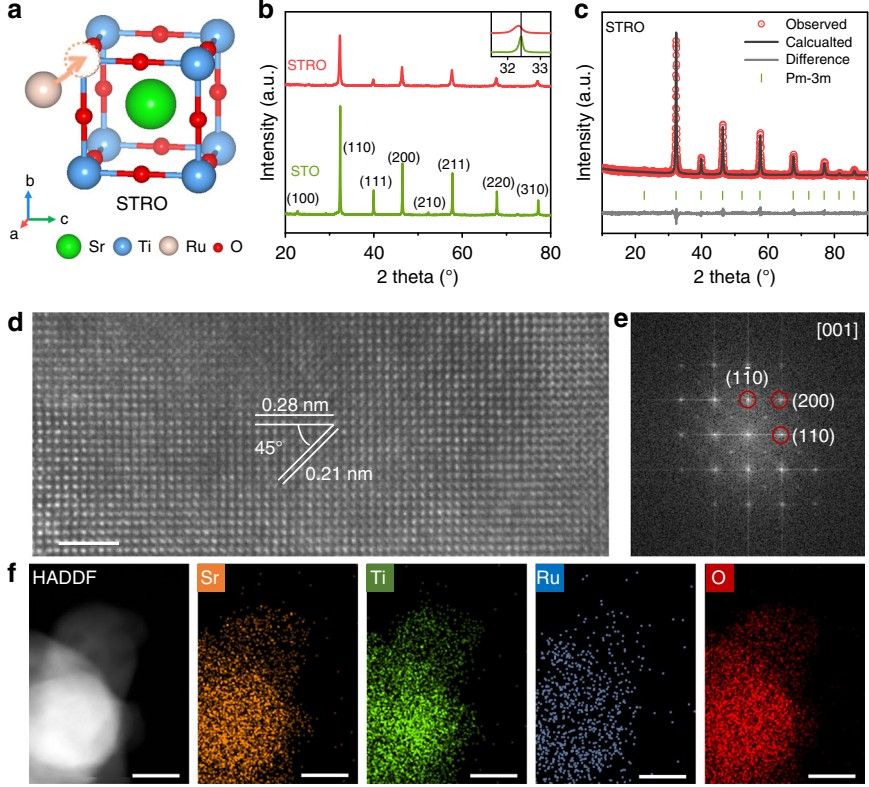

**Fig. 2 Structural characterization of STRO perovskite. a** Schematic presentation of STRO perovskite structure. **b** XRD patterns of STO and STRO. Inset is the expanded region of main peaks. **c** Refined XRD profile of the STRO. **d** HRTEM image of STRO, and **e** the corresponding FFT pattern. **f** HAADF-STEM and the corresponding elemental mapping images of STRO. Scale bar in **d** is 2 nm and in **f** is 20 nm.

STRO perovskite was discovered, which contributes to the 180° interaction between neighboring $Ti^{3+}$ (with a $3d^1$ configuration) and $Ru^{5+}$ (with a $4d^3$ configuration) ions according to the Goodenough–Karamori–Anderson rule, and enhanced electrical conductivity. The STRO shows an exceptional HER activity with a low overpotential of 46 mV at 10 mA cm$^{-2}$ and a very small Tafel slope of 40 mV dec$^{-1}$ in 1 M KOH, comparable to benchmark Pt/C catalyst and surpassing most state-of-the-art catalysts ever reported. In addition, the STRO demonstrates robust operational stability up to 200 h in alkaline HER condition. By density functional theory (DFT) calculations, the excellent activity is primarily attributed to a unique synergistic effect among multiple catalytic sites in STRO; that is, the Ti ions serves in accelerating water dissociation, Ru ions favor OH* desorption, and non-metal oxygen sites (i.e., oxygen vacancies/lattice oxygen) are active for the nearly optimal H* adsorption and $H_2$ evolution. This work brings fundamental insights into the role of single-phase multi-active-site synergy and provides a promising way for developing advanced alkaline HER electrocatalysts via an atomic-level modulation strategy.

## Results

**Synthesis and structural characterization of STRO.** The Ru-doped $SrTiO_3$ perovskite oxides with different Ti/Ru molar ratios ($SrTi_{1-x}Ru_xO_{3-\delta}$, $x = 0$, 0.1, 0.2, 0.3, 0.4, 1) were synthesized by a facile and scalable solid-state reaction method (see the experimental section for details on samples synthesis). Among all $SrTi_{1-x}Ru_xO_{3-\delta}$ samples with different compositions and calcination temperatures, the $SrTi_{0.7}Ru_{0.3}O_{3-\delta}$ calcined at 900 °C shows the best HER activity (Supplementary Figs. 1–4), and hence $SrTi_{0.7}Ru_{0.3}O_{3-\delta}$ was chosen as the model catalyst for study in this

work. For clarity, the $SrTi_{0.7}Ru_{0.3}O_{3-\delta}$, $SrTiO_3$, and $SrRuO_3$ are denoted as STRO, STO, and SRO hereafter, respectively. In the ideal cubic-symmetry perovskite structure, larger A-site cations (e.g., Sr) are 12-fold oxygen coordination, and smaller B-site cations (e.g., Ti/Ru) are sixfold oxygen coordinated (Fig. 2a). X-ray diffraction (XRD) was initially used to verify the phase structure. As shown in Fig. 2b, all the diffraction peaks for the STO and STRO can be well indexed as a cubic perovskite structure without the appearance of any impurity phase. For the STRO, the main peaks shift slightly to lower angles as compared with those for the pristine STO, implying a lattice expansion after the Ru doping. Additionally, Rietveld refinement of the XRD pattern reveals that the STRO possesses a pure cubic phase with a space group of Pm-3m and unit cell parameters of a = 3.9178 Å (Fig. 2c and Supplementary Table 1). The crystal structure of STRO was further confirmed by the high-resolution transmission electron microscopy (HRTEM) image and corresponding fast Fourier transform (FFT) image along the [001] direction. In Fig. 2d, there are two different lattice fringes with interplanar distances of 0.28 nm and 0.21 nm, corresponding to the (110) and (200) crystal planes of cubic STRO; meanwhile, FFT pattern also supports the cubic phase (Fig. 2e). Furthermore, the high-angle annular dark-field scanning transmission electron microscopy (HAADF-STEM) and elemental mapping images demonstrate the homogeneous distribution of all elements in the as-prepared STRO material (Fig. 2f). The surface morphology was examined by scanning-electron microscopy (SEM). Similar aggregates composed of nanometer-sized particles were observed in STO, STRO, and SRO samples (Supplementary Figs. 5 and 6), which suggests the nature of bulk materials and the surface morphology would not contribute to their distinct electrocatalytic activities, as to be discussed below.

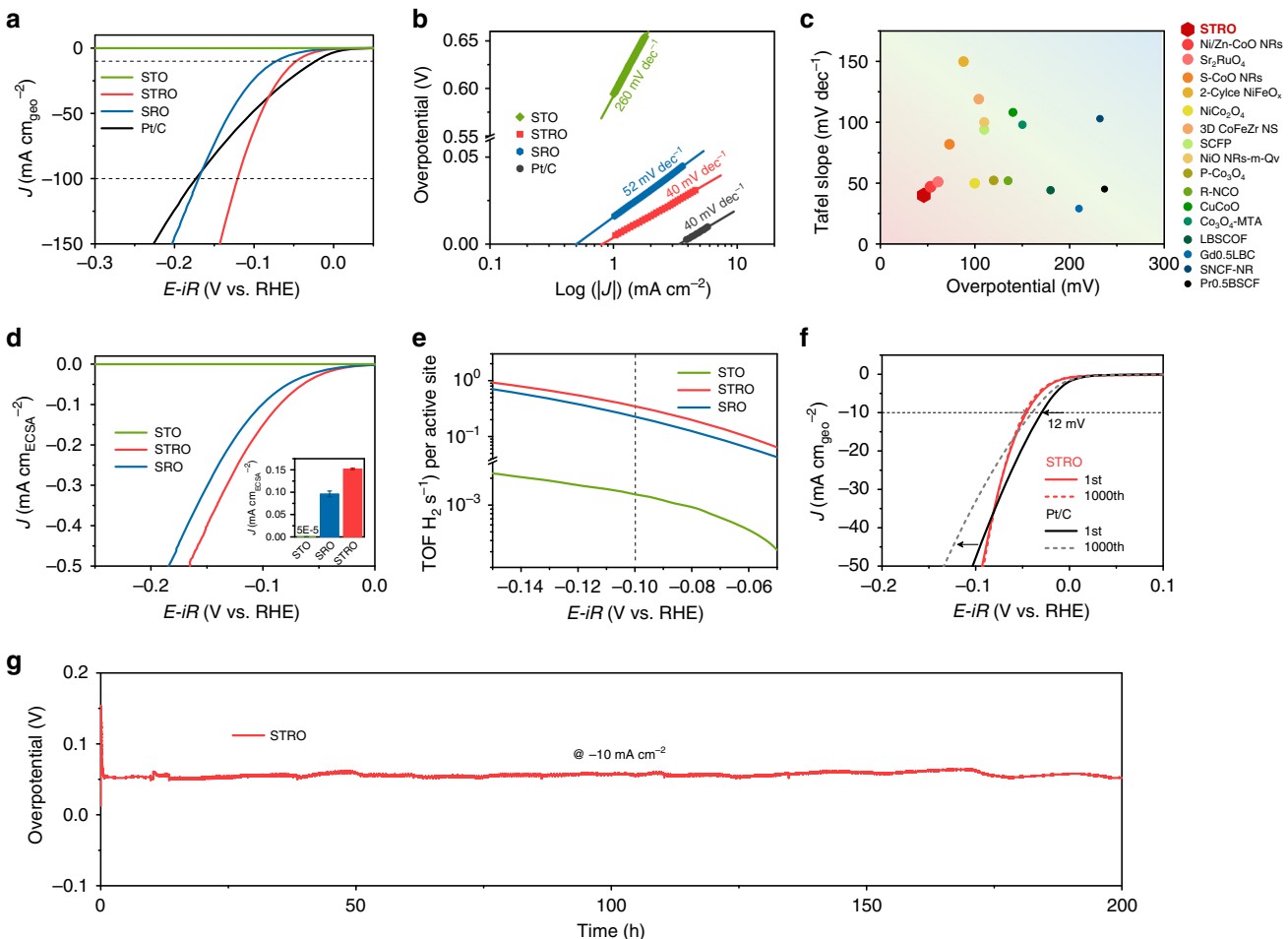

**Fig. 3 Electrocatalytic HER performance in alkaline media. a** Polarization curves of STO, STRO, SRO, and commercial Pt/C catalysts in an Ar-saturated 1 M KOH solution. Scan rate, 5 mV s⁻¹. **b** The corresponding Tafel plots. **c** Alkaline HER activity comparison among metal oxides involving the overpotential@-10 mA cm⁻² and Tafel slope. **d** Specific activity normalized to ECSA of STO, STRO, and SRO catalysts as a function of applied potential. Inset: specific activity at the overpotential of $\eta = 0.1$ V. **e** The relationship between TOF and the tested potentials for STO, STRO, and SRO catalysts in 1 M KOH solution. **f** Polarization curves of STRO and commercial Pt/C catalyst initially and after 1000 cycles during the accelerated durability tests. **g** Chronopotentiometry response of STRO at a constant cathodic current density of 10 mA cm⁻².

**Comprehensive evaluation of HER performance**. The alkaline HER electrocatalytic performance of the STO, STRO, and SRO catalysts was evaluated on the three-electrode configuration in 1 M KOH solution. For comparison, similar measurements were also carried out on the benchmark Pt/C catalysts. If not specified otherwise, all potentials in this work were *iR*-corrected to remove the ohmic drop across the electrolyte and referenced to a reversible hydrogen electrode (RHE, see Supplementary Fig. 7 for calibration). Figure 3a shows the polarization curves obtained from linear sweep voltammetry (LSV) at a scan rate of 5 mV s⁻¹. STRO exhibits a much smaller onset overpotential (defined here as the overpotential at 1 mA cm⁻²) of ~4 mV (close to that of Pt/C) and greater catalytic current than STO and SRO, suggesting significantly enhanced alkaline HER activity after proper Ru doping. The negligible catalytic current was observed for pristine STO, implying its intrinsic inertness for HER. Noticeably, the HER current of STRO can largely exceed that of the benchmark Pt/C catalyst beyond −0.08 V, which can be ascribed to the underwater superaerophobic surface of STRO. The super-aerophobic surface of STRO, as reflected by the high contact angle of the gas bubble (Supplementary Fig. 8), can promote the quick leaving of as-generated H₂ bubbles and facilitate mass-transport, especially at large current densities[32,33]. Besides, the STRO delivers a small overpotential of 46 mV at −10 mA cm⁻²,

much lower than that of STO and SRO, and even close to that of benchmark Pt/C. To examine the kinetics, Tafel plots were drawn in Fig. 3b. The Tafel slope for STRO (40 mV dec⁻¹) is smaller than that for STO (260 mV dec⁻¹) and SRO (52 mV dec⁻¹), implying faster HER rates. Moreover, the exchange current density ($j_0$) was also obtained from extrapolating the Tafel plots to zero overpotential, which reflects the intrinsic electron transfer ability during electrocatalysis[34]. STRO offers a $j_0$ value of 0.805 mA cm⁻², ~twofold higher than SRO, and markedly two orders of magnitude higher than STO. Overall, the above electrochemical analyses (e.g., small overpotential, low Tafel slope, and high exchange current density) highlight the extraordinary catalytic activity of STRO for HER in alkaline media. Such excellent HER activity of STRO is comparable to those state-of-the-art metal oxides (Fig. 3c) and synergistic hybrids as well as various representative catalysts reported to date (Supplementary Table 2), demonstrating that STRO is among the most active alkaline HER catalyst.

To assess the intrinsically catalytic activity of each active site, we further calculated the specific activity by normalizing the electrode activity to the real oxide surface area (ROSA) and electrochemical surface area (ECSA). The values of ROSA and ECSA of catalysts were estimated from the BET method (Supplementary Fig. 6) and double-layer capacitance ($C_{dl}$)

measurements (Supplementary Fig. 9), respectively. In Fig. 3d and Supplementary Fig. 10, the STRO catalyst presents a much higher specific activity than STO and SRO, regardless of the current densities normalized to the ROSA or ECSA. For example, STRO gives a specific activity of $0.15 \, \text{mA cm}^{-2}_{\text{ECSA}}$, which is ~3000- and ~1.5-fold higher than that for STO and SRO at $\eta = 0.1 \, \text{V}$. Furthermore, the turnover frequency (TOF), relevant to the amount of gaseous hydrogen molecule evolving per active oxygen site per second, is another crucial parameter reflecting the intrinsic activity of an electrocatalyst[35,36]. Here, the well-established method reported by the Jaramillo group was adopted to calculate the TOF values (see Supplementary Fig. 11 and Supplementary Note 1)[35,36]. Figure 3e shows the TOF versus potential plots of STO, STRO, and SRO catalysts, displaying that TOF values follow the order in the sequence of STRO > SRO >> STO. These results indicate that STRO possesses higher intrinsic activity than STO and SRO, highlighting the significant promoting role of Ru/Ti dual metals. Furthermore, the mass activity (MA) and price activity (PA) of SRO, STRO, and Pt/C was also calculated (Supplementary Fig. 12). It can be seen that the STRO catalyst exhibits both much higher mass activity and price activity than the commercial Pt/C catalyst, demonstrating the cost-effectiveness of STRO catalyst in practical applications. In addition to catalytic activity, long-term operation stability is another critical criterion to evaluate the potential of catalysts for practical application. To this end, the accelerated durability tests (ADT) by continuous cycling within HER potential window were conducted. After 1000-cycling, the STRO exhibits an almost identical polarization curve with the initial one, indicative of its good durability (Fig. 3f). In contrast, benchmark Pt/C suffers from obvious activity decay with an overpotential increase of ~12 mV after the ADT measurement. Moreover, the negligible fluctuation of overpotential (@ −10 mA cm$^{-2}$) was observed during 200 h chronopotentiometry test (Fig. 3g), which further confirms the robust stability during long-term operation. Further evidence to support the stability of STRO during HER is from XRD, X-ray absorption spectra (XAS) and TEM analyses. Neither change in XRD and XAS peaks and nor surface amorphization in TEM images was observed for STRO catalyst before and after 1000-cycle ADT (Supplementary Figs. 13–15), demonstrating that STRO is stable without structure reconstructions under alkaline HER condition. Overall, in terms of activity and stability, STRO as an excellent electrocatalyst holds great promise for practical application in alkaline water electrolysis.

**Charge redistribution** via **super-exchange interaction**. To gain some insight into the origin of the high HER activity of STRO, we explored the valence state and electronic structure information using the synchrotron-based X-ray absorption spectra (XAS) technique. Figure 4a depicts the Ti $L$-edge XAS spectra of STO and STRO comprising of $L_2$ and $L_3$ doublets, which are caused by the electron excitation from Ti $2p_{1/2}$ and $2p_{3/2}$ to unoccupied 3d orbitals, respectively. As a result of the crystal-field splitting in octahedral symmetry, both of these doublets split into $e_g$ and $t_{2g}$ peaks[37]. The intra-atomic multiplet interaction leads to two very weak pre-edge peaks at 455.0 eV and 455.7 eV. In the enlarged spectrum of the Ti-$L_3$ edge (Fig. 4b), a lower-energy shift and broader asymmetry $e_g$ peak were observed for STRO as compared to pristine STO, implying the reduction of Ti valence state after low-level Ru doping. Thus, the incorporation of Ru dopants in STO leads to an increased electron density around Ti$^{4+}$ sites and the generation of Ti$^{3+}$ sites[38]. Besides, the Ru $L_{2,3}$-edge XAS spectra of SRO and STRO were also shown in Fig. 4c. As seen from Fig. 4d, an obvious positive shift of Ru-$L_3$ spectrum to the higher energy of ~0.5 eV is visible for STRO relative to SRO,

indicating an increased valence state of Ru ions from Ru$^{4+}$ to Ru$^{5+}$[39,40]. Bader charge analysis was further performed to study charge density differences. As shown in Fig. 4e and Supplementary Table 3, the charge of Ti and Ru atoms in STRO is calculated to be +2.10 |e| and +1.84 |e|, compared to +2.20 |e| on Ti atoms in STO and +1.51 |e| on Ru atoms in SRO, respectively, suggesting the electron donation from Ru to Ti in STRO, in accordance with the XAS results. Considering the variation of the Ti and Ru valence states within STRO, a charge redistribution between Ti and Ru ions (Ti$^{4+}$+Ru$^{4+}$→Ti$^{3+}$+Ru$^{5+}$) is deduced and gives rise to an unusual super-exchange interactions between adjacent Ti (III) and Ru (V) sites[30,41,42], which is schematically illustrated in Fig. 4f.

**Enhanced oxygen vacancy and electrical conductivity**. In addition to the generation of the Ti$^{3+}$ and Ru$^{5+}$ ions via the super-exchange effect, STRO also shows the enhancement in another two key factors compared to the pristine STO as follows: (1) *oxygen vacancies (OVs)*: According to the charge neutrality and iodometric titration, the oxygen non-stoichiometry in STRO is calculated to be ~0.2 (Fig. 5a), implying the existence of OVs. The increased OVs in STRO were further probed by the X-ray photoelectron spectroscopy (XPS) and electron paramagnetic resonance (EPR) techniques. XPS spectra of O 1 s species (Fig. 5b) and the corresponding deconvolution results (Supplementary Table 4) demonstrate a much larger number of (O$_2^{2-}$/O$^-$) species, which is closely related to the surface oxygen vacancies[30,43,44], in STRO relative to STO, suggesting the increased OVs. EPR spectra were recorded to provide fingerprint evidence due to its sensitivity to unpaired electrons trapped by oxygen vacancies[45,46]. A strong signal intensity at $g = 2.004$ was observed for STRO, while STO has a very weak signal (Fig. 5c), revealing a much higher concentration of OVs in STRO. (2) *Electrical conductivity*: As known, the electrocatalysis process requires an efficient flow of electrons through the electrode to produce high currents; thus, catalysts with high electrical conductivity are beneficial to high-efficiency electrocatalysis[47,48]. First, the enhanced electrical conductivity of STRO (~0.2 S cm$^{-1}$) (vs. STO with ~0 S cm$^{-1}$) at room temperature was confirmed by the four-probe direct current (DC) measurement (Fig. 5d). Furthermore, the UV–vis absorption spectra for STO and STRO (Supplementary Fig. 16) were performed to explore the electronic behavior, and the corresponding optical band gaps were calculated according to the Tauc equation (Fig. 5e). The bandgap of STO largely narrows from 3.20 eV to 2.20 eV after introducing Ru dopants into its lattice matrix, which favors the excitation of charge carriers to the conduction band and leads to an enhanced electrical conductivity of STRO[49,50]. The electronic properties can be also revealed from the profiles of the density of states (DOS). As can be seen in Fig. 5f, the pristine STO is a semiconductor with a large bandgap[40]. However, after Ru doping, the STRO presents a metallic characteristic with predominantly Ru 4d orbital crossing the Fermi level, which accounts for the increased conductivity. Based on above-combined analyses, the oxygen vacancy generation and electrical conductivity enhancement in STRO are demonstrated, which are expected to promote the HER catalytic activity.

**DFT calculations and electrocatalysis mechanism**. To further understand the underlying alkaline HER mechanism on STRO perovskite oxide, we resorted to first-principle density functional theory (DFT) calculations for both kinetic and thermodynamic aspects. We used the BO$_2$-terminated (001) plane to model STRO surface in our calculations, considering that it is usually the most observed and most stable termination for cubic-symmetry

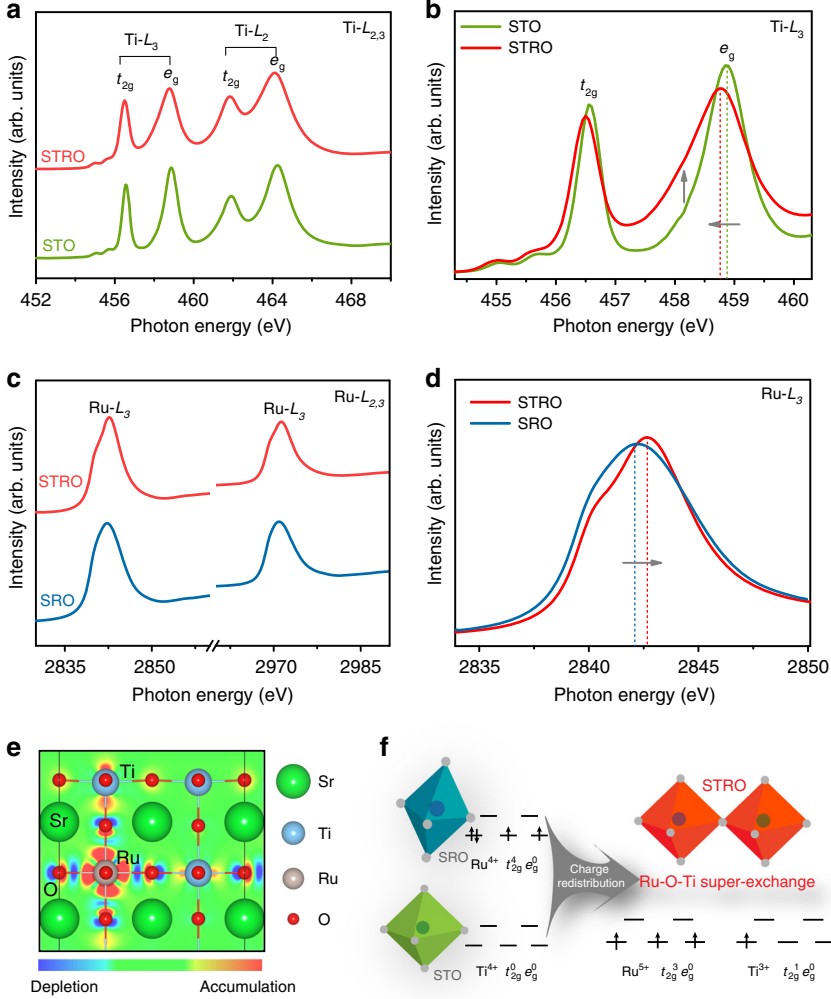

**Fig. 4 Charge redistribution via super-exchange interaction. a** Normalized Ti $L_{2,3}$-edge XAS spectra of STO and STRO. **b** Enlargement of the Ti $L_3$-edge XAS spectra of STO and STRO. **c** Normalized Ru $L_{2,3}$-edge XAS spectra of STRO and SRO. **d** Enlargement of the Ru $L_3$-edge XAS spectra of STRO and SRO. **e** Top view of the charge distribution in STRO. **f** Schematic illustration of the super-exchange interaction in STRO.

perovskites, as revealed from both experimental and theoretical studies[40,51]. As stated before, different from the acid HER, the more complex reaction pathways under alkaline condition involve several key elementary steps, including prerequisite water dissociation to adsorbed H* and OH⁻, subsequent OH⁻ desorption from the surface, and the concomitant combination of adsorbed H* into molecular H₂. Thus, in terms of kinetics, DFT calculations were first utilized to investigate the kinetic reaction barriers ($E_b$) for breaking the OH–H bond. The surface structure configurations at the initial state (IS) and final state (FS) of STO, STRO, and SRO for the initial water-dissociation process is shown in Fig. 6a, and the corresponding $E_b$ values are calculated. As shown in Fig. 6b, STO and SRO exhibit relatively good kinetics for water dissociation with $E_b$ of 0.17 and 0.32 eV, respectively. Such low water-dissociation barriers in STO are in good agreement with previous studies, which reveals facile kinetics for water dissociation[52]. Strikingly, a very low $E_b$ value of 0.03 eV was calculated at the Ti sites of STRO, significantly lower than that at the Ru sites (0.38 eV). The negligible energy barrier for breaking the OH–H bond demonstrates that the water-dissociation step in STRO is energetically favorable on Ti sites.

In addition, the free energy diagram of alkaline HER process was constructed to gain further insight of the active sites from a thermodynamic viewpoint. As shown in Fig. 6c, the negative free energy indicates an exothermic Volmer step for all catalysts, revealing that water dissociation on the surface of these perovskite oxides is thermodynamically favorable. Following the water-dissociation step, the generated OH⁻ ions need to desorb from the catalyst surface for the regeneration of active sites, and the H* species are adsorbed on the active sites for the subsequent H₂ recombination. Consequently, we also calculated the other two key activity parameters: the adsorption free energy of H* ($\Delta G_{H*}$) and desorption free energy of OH* ($\Delta G_{OH*}$). The Ti sites on the surface of STO and STRO both display large positive $\Delta G_{OH*}$ values, indicative of a difficult OH* desorption, which may be due to the highly unfilled d orbitals of Ti³⁺/⁴⁺ ions with excessively strong electrostatic attraction tendency to immobilize the OH⁻[14]. On contrast, the OH* desorption at Ru sites on the surface of SRO and STRO is remarkably facilitated with lower $\Delta G_{OH*}$ values of −0.42 eV and 0.29 eV, respectively. Hence, Ru sites on the surface of STRO with moderate OH* binding energy can facilitate the OH⁻ desorption. Moreover, $\Delta G_{H*}$ is usually regarded as an important descriptor to assess the HER activity, and a $\Delta G_{H*}$ value close to zero leads to optimal HER activity owing to the optimal balance between H* absorption and desorption[4,6,7,14,17]. The $\Delta G_{H*}$ values at lattice-oxygen sites for STO and SRO are −0.33 eV and −0.69 eV, respectively, indicating that the H* adsorption is too strong. Nevertheless, two non-metal sites (namely, lattice-

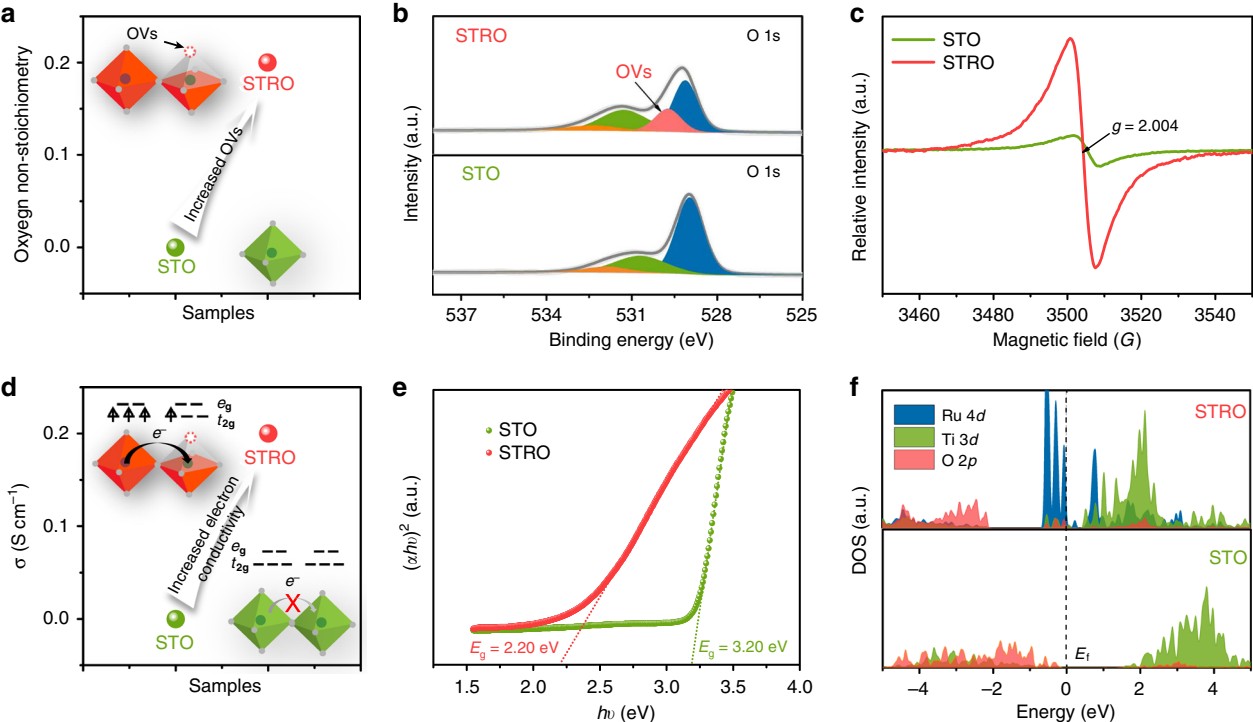

**Fig. 5 Enhanced oxygen vacancy and electrical conductivity. a** The oxygen non-stoichiometry of STO and STRO. **b** O 1s XPS spectra of STO and STRO. **c** EPR spectra of STO and STRO. **d** The electronic conductivity of STO and STRO at room temperature. **e** Tauc plots of UV–vis spectroscopy for STO and STRO. **f** DOS profiles of STO and STRO.

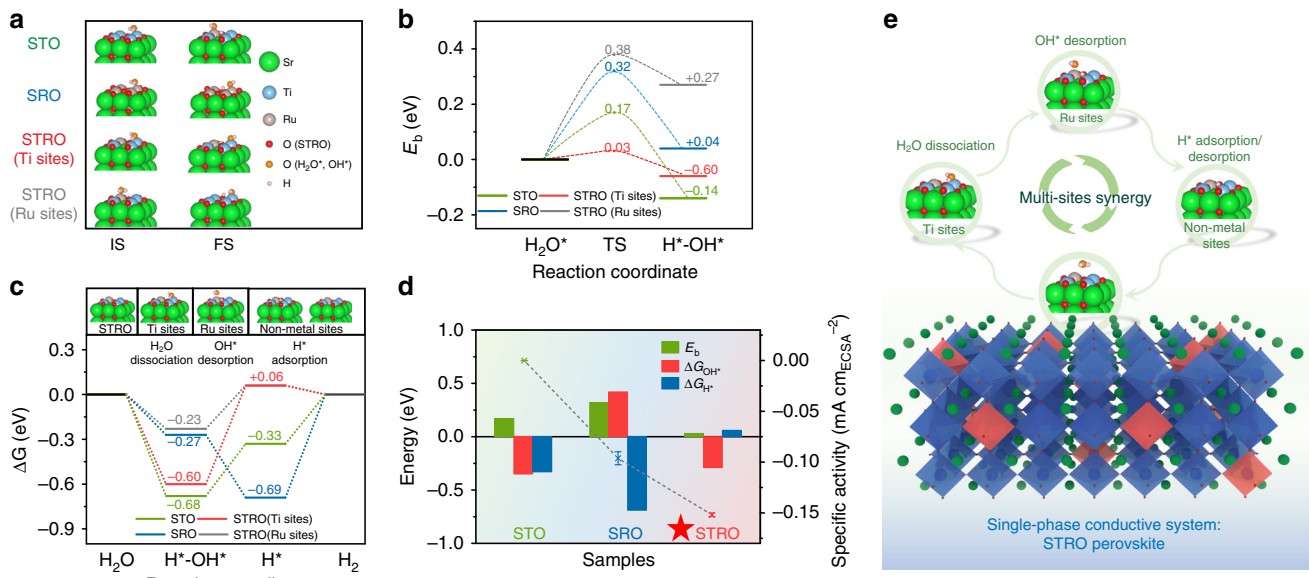

**Fig. 6 DFT calculations and electrocatalysis mechanism. a** Surface structure configurations at the IS ($H_2O$) and FS ($H^* + OH^*$) of STO, STRO, and SRO in the prior water-dissociation process. **b** The energy barrier for breaking the OH–H bond at the transition state (TS) in the water-dissociation process. **c** Gibbs free energy diagrams of the alkaline HER pathway on STO, STRO, and SRO. The inset above is the surface configuration of STRO at different stages of the reaction. **d** The relationship between the computed $E_b$, $\Delta G_{H^*}$, $\Delta G_{OH^*}$ values, and the measured activity on STO, STRO, and SRO catalysts. **e** Schematic illustration of synergistic catalysis mechanism for alkaline HER on the single-phase STRO perovskite oxide. The red and blue octahedra represent $RuO_6$ and $TiO_6$, respectively.

oxygen and oxygen vacancy) in STRO possess optimal H* adsorption strength. Impressively, $\Delta G_{H^*}$ at the lattice-oxygen and oxygen vacancy sites of STRO are only 0.06 and 0.05 eV, respectively, which are both very close to the ideal zero (Supplementary Fig. 17). In order to further investigate the influence of oxygen vacancy on the activity, we prepared a $SrTiO_{3-\delta}$ perovskite with the generation of oxygen vacancies via a reductive atmosphere treatment (denoted as R-STO, see Supplementary Figs. 18–20 for more sample details). As can been seen, R-STO shows significantly higher activity than pristine STO

(Supplementary Fig. 21), demonstrating the positive role of oxygen vacancies in promoting HER. Besides, the enhanced super-exchange interaction in R-STO may also contribute to increased activity.

Taken together, three important parameters (i.e., $E_b$, $\Delta G_{H*}$ and $\Delta G_{OH*}$) can all affect the alkaline HER activity separately; however, none of them alone can act as a single descriptor for predicting the activity trend (Supplementary Fig. 22). Instead, these parameters collectively determine the overall reaction rate, leading to the experimentally observed activity trend, where they are optimum for STRO (Fig. 6d). To sum up, the HER mechanism on STRO perovskite oxide in alkaline media via multi-active-site synergy was disclosed, as schematically illustrated in Fig. 6e. In other words, the synergistic active centers, including facile water dissociation at Ti sites, favorable $OH^-$ desorption at Ru sites and optimal $H^*$ adsorption at non-metal oxygen sites (i.e., oxygen vacancies/lattice oxygen), together with high electrical conductivity, collectively contribute to the exceptional alkaline activity of STRO (Supplementary Table 5).

Inspired by the key role of super-exchange behavior between Ru and Ti in STRO system in promoting alkaline HER activity, we were intrigued to find out whether other dopants into STO can also possess a similar phenomena. First, we performed additional charge analysis for several other metal dopants on the surface of STO, e.g., Ir, Mo, Nb, and Pt (Supplementary Fig. 23). It was found that the charges on Ti are 2.10, 2.04, 2.03, and 2.05 $|e|$ for Ir, Mo, Nb, and Pt dopants, respectively, which are very similar to the case $(2.10 \,|\, e \,|)$ of Ru dopant, suggesting a charge transfer from the metal dopants to the surface Ti sites. This result could be a clear signal of the presence of $Ti^{3+}$ along with $M^{5+}$ surface species. To further experimentally confirm the positive role of $Ti^{3+}/M^{5+}$ exchange behavior in alkaline HER activity, we tried to prepare one Mo-doped $SrTi_{0.7}Mo_{0.3}O_{3-\delta}$ perovskite (denoted as STMO) and compared its activity with STO and $SrMoO_4$ (SMO) (Supplementary Figs. 24 and 25). STMO exhibits higher catalytic activity than STO and SMO, suggesting the super-exchange behavior between Mo and Ti (similar to STRO) can also boost alkaline HER. Based on the above theoretical and experimental results, we believe that constructing $Ti^{3+}/M^{5+}$ couples in perovskites via super-exchange effect may be a universal way for boosting alkaline HER.

## Discussion

In summary, a perovskite oxide with unusual super-exchange effect, $SrTi_{0.7}Ru_{0.3}O_{3-\delta}$ (STRO), has been developed as a highly active and durable HER electrocatalyst in alkaline media. When evaluated in 1 M KOH solution, STRO displays prominent HER activity with a low overpotential of only 46 mV at 10 mA cm$^{-2}$ and a small Tafel slope of 40 mV dec$^{-1}$, which is at the top level in terms of catalytic activity among all state-of-the-art catalysts ever reported and even comparable to the benchmark Pt/C catalyst. Moreover, the STRO catalyst possesses excellent stability during long-time HER operations. Combined experimental and theoretical studies reveal that the high catalytic activity of STRO results from the creation of synergistic active sites and enhanced electrical conductivity, which is induced by the charge redistribution via $Ti^{3+}$-O-$Ru^{5+}$ super-exchange interactions. Through DFT calculations, the near-optimal synergistic interplay of alkaline HER intermediates among multiple catalytic sites in STRO is uncovered: the Ti sites boost water dissociation with a negligible kinetic barrier, Ru sites are favorable for $OH^*$ desorption, and non-metal oxygen sites (i.e., oxygen vacancies/lattice oxygen) serve as the locations for the nearly optimal $H^*$ adsorption and $H_2$ desorption; thus accelerating the overall alkaline hydrogen-

evolution process. The proof-of-concept study not only reports a single-phase perovskite oxide with exceptional HER catalytic performance but also proposes a concept for designing advanced electrocatalysts for other applications via constructing atomic-scale synergistic active sites.

## Methods

**Catalyst synthesis**. $SrTi_{1-x}Ru_xO_{3-\delta}$ ($x = 0, 0.1, 0.2, 0.3, 0.4, 1$) were prepared through the traditional solid-phase reaction route. Taking the synthesis of $SrTi_{0.7}Ru_{0.3}O_{3-\delta}$ (STRO) as an example, stoichiometric amounts of $SrCO_3$, $TiO_2$, and $RuO_2$ were weighed and mixed in ethanol under the rotation speed of 400 rpm for 1 h via a high-energy ball mill (Planetary Mono Mill, Pulverisette 6, Fritsch). The homogeneously dispersed mixture was then dried and calcined at different temperatures in air for 5 h to form the resultant powders. Commercial Pt/C catalyst was purchased from Johnson Matthey Company.

**Characterizations**. XRD patterns were measured using a Rigaku Smartlab diffractometer operating at 40 kV with filtered Cu Kα radiation. The Rietveld refinements were revealed using DIFFRAC plus Topas 4.2 software. SEM images were recorded through a scanning-electron microscope equipped with the scanning-electron microanalyzer (Hitachi S-4800). The HRTEM images were obtained utilizing the electron microscope (FEI Tecnai G2 F20) operating at 200 kV. STEM image and elemental mapping images were obtained using Tecnai F20 SuperTwin operating at 200 kV. Nitrogen adsorption–desorption isotherms were recorded on BELSORP II. O 1 s spectra were acquired on X-ray photoelectron spectroscopy (Perkin Elmer PHI 1600 ECSA system) and fitted through the XPSPEAK software package. EPR spectra were obtained using a Bruker EPR A300 spectrometer. XAS spectra of Ti-$L$ edge and Ru-$L$ edge were determined at the BL 11A and BL 16A beamline of the National Synchrotron Radiation Research Center (NSRRC) in Taiwan. All samples were pretreated via cutting pellets in an ultrahigh vacuum chamber to obtain a clean surface. $Sr_2GdRuO_6$ and $Sr_2RuO_4$ single crystals were measured to calibrate the energy scale. UV–Vis absorption spectra were investigated by a spectrophotometer (HITACHI U-3010). The electrical conductivity at room temperature for STRO and parent STO was measured through the four-probe direct current (DC) technique in an air atmosphere. In addition, the voltage and the current signals were taken by utilizing a Keithley 2420 source meter.

**Electrochemical measurements**. HER measurements in alkaline media were conducted in a standard three-electrode electrochemical cell (Pine Research Instrumentation) in an RDE configuration using a CHI 760E electrochemistry workstation. Catalysts cast on RDE (5 mm in diameter), graphite rod, and Hg|HgO (1 M KOH) were used as the working electrode, counter electrode, and reference electrode, respectively. Working electrodes for HER measurements were prepared by a controlled drop-casting method, which is in accordance with the previous works[4,29,30]. The mass loading of perovskite catalysts on the RDE is ~0.232 mg$_{oxide}$ cm$^{-2}$. Linear sweep voltammetry (LSV) was recorded at 5 mV s$^{-1}$ at the rotation of 2400 rpm in Ar-saturated 1 M KOH to obtain the HER-polarization curves. Tafel slopes were determined by plotting the overpotential versus the logarithm of current density (log$\,|\,J\,|$). The exchange current densities were calculated by extrapolating the Tafel plots to the overpotential of 0 V. Cyclic voltammetry with the continuous potential cycling between −0.8 V and −1.4 V vs. Hg|HgO at 100 mV s$^{-1}$ and chronopotentiometric tests at the current density of −10 mA cm$^{-2}$ were conducted to explore the stability of the electrocatalysts. The electrochemical double-layer capacitance ($C_{dl}$) was measured through the CV technique within the potential window from −0.843 to −0.743 V vs. Hg|HgO of the non-faradic current region at the changed sweeping rates from 20 to 100 mV s$^{-1}$. The $C_{dl}$ values were determined by plotting the halves of the differences between positive and negative current density versus the scan rate.

**Computational methods**. Electronic structure calculations were performed using the Vienna ab initio Simulation Package (VASP version 5.4.4)[53,54]. Core and valence electrons were treated within the projector augmented wave (PAW) framework[55], with wavefunctions expanded up to a kinetic energy cutoff of 500 eV. Exchange and correlation were described using the RPBE functional[56]. Spin polarization was always accounted for. STO and SRO surfaces were cleaved from their respective cubic and orthorhombic phases, where the (001) $BO_2$ termination was used to model the catalytic reactions. A symmetric $2 \times 2$ surface with nine atomic layers was used, which is thick enough to prevent spurious interactions across the perovskite. A vacuum layer of >18 Å was added along the $z$ direction to minimize periodic interactions. Intermediates' free energies of adsorption were calculated by considering the binding energies along with the entropic and zero-point energy corrections. Such calculations were performed within the context of the computational standard hydrogen electrode framework developed by Nørskov and coworkers[57,58], and were used to construct the reaction energy profiles for the alkaline HER (see the Supplementary Note 2 for more details). Water-dissociation barriers were calculated using the climbing image nudged elastic band method[59].

## Data availability

The data that support the findings of this study are available from the corresponding authors upon request.

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

## Acknowledgements

This work was financially supported by National Natural Science Foundation of China of Nos. 21576135, 21706129, and 21878158, Jiangsu Natural Science Foundation for Distinguished Young Scholars of No. BK20170043. The authors acknowledge the support from the Max Planck-POSTECH-Hsinchu Center for Complex Phase Materials, the DFG through SFB 1143, the resources provided by the Pawsey Supercomputing Centre with funding from the Australian Government and the Government of Western Australia, and the use of facilities within the Monash Centre for Electron Microscopy. Dr. Y. Zhu acknowledges the Australian Research Council (Discovery Early Career Researcher Award No. DE190100005).

## Author contributions

Y.L.Z. and Z.P.S. conceived and designed the research. J.D. conducted characterizations and electrochemical measurements. H.A.T. and S.C.S. performed DFT calculations. Q.L. was involved in the structural and electrochemical analysis. Y.C. performed TEM characterizations. D.Q.G. performed XRD refinements. D.Q.G., Z.W.H., H.J.L., T.S.C., and C.T.C. performed XAS characterizations. C.Z. performed conductivity measurements. All authors discussed and analyzed the data. D.J., Y.L.Z., H.A.T., H.T.W., W.Z., and Z.P.S. co-wrote the paper.

## Competing interests

The authors declare no competing interests.
