## [Peer Review File · Nature Communications]

REVIEWER COMMENTS

Reviewer #1 (Remarks to the Author):

This work presents a very active and novel electrocatalyst for the HER. The primary claim of novelty is that their STRO material acts as a single-phase synergistic electrocatalyst. The activity of this electrocatalyst is the highest of any perovskite electrocatalyst in the literature and amongst the highest non-precious metal electrocatalysts reported. The data supporting this claim are very strong and this catalyst does appear to be very exciting. I can see no issue with the experimental approaches used to collect this data and am impressed with the breadth of analysis used. I believe this paper contributes not only a very good electrocatalyst, but also a strong mechanistic and theoretical basis for this activity. The authors describe a new method through which one can optimise a perovskite electrocatalyst - that of synergistic coupling of two B-site cations and lattice oxygen / vacancies. For these reasons, I recommend publication of the manuscript in Nature Communications with some changes noted below.

1. Page 4 (lines 85-88): The binding of OH⁻ should be optimal (neither too weak nor too strong), not necessarily as weak as possible as implied by the highlighted lines.

Dubouis, N. & Grimaud, A. The hydrogen evolution reaction: From material to interfacial descriptors. *Chem. Sci.* 10, 9165–9181 (2019)

2. At the top of page 5 it is implied that this is a cost-effective catalyst. However, there is no cost comparison in the paper to Pt or other high-activity non-PGM electrocatalysts. Given that the reported catalyst has activity close to that of platinum, it is highly likely that including a cost-based metric would make this material even more impressive.

Zhu, Y. L. et al. Unusual synergistic effect in layered Ruddlesden - Popper oxide enables ultrafast hydrogen evolution. *Nat. Commun.* 10, 9 (2019)

3. The Tafel plot of the Pt/C electrocatalyst would be interesting as a comparison in Figure 3b. Generally, given that Pt/C is the current catalyst of choice, it would be worth including it in D and E as well.

4. The arrow on Figure 3c is unnecessary and the trend it indicates appears to be rather weak.

5. The figure caption for 3g states a 'cathodic current density of -10 mA cm⁻²'. A cathodic current density by definition (at least according to IUPAC) is negative, so the minus sign is redundant.

6. Does the BET surface area give any useful activity? The BET measurement is made on the dry catalyst, whereas the electrocatalyst ink is a mixture of the perovskite, conductive carbon, Nafion. This would presumably greatly change the accessible surface area of the electrode. As such supplementary figure 9 which is normalised to the ECSA would perhaps be of more relevance than Figure 3d?

7. There are no mass-normalised current densities presented in the work. In addition to current densities at given overpotentials, it is useful to provide mass activity (A g⁻¹) to increase comparability with other materials.

8. The figure legend in supplementary figure 9 has labelled all the lines wrong.

9. The downward arrow in Figure 6b doesn't add anything to the figure and makes reading the plot harder.

10. The red and blue octahedra in Figure 6e should be addressed in the figure caption. Presumably the red octahedra refer to RuO₆?

11. In the experimental section you should state the size/area of the working electrode to aid reproducibility.

12. A general note of the figures: The large half-filled circles used in many of the plots makes reading the data and seeing the error bars difficult. A small cross or similar would improve readability

13. Generally, this paper is well-written but there are several instances of poor/inaccurate grammar and wording which should be caught with a thorough proof-reading.

Overall, this is an excellent contribution to the literature and will be of significant interest to those working in electrocatalysis.

Reviewer #2 (Remarks to the Author):

The authors investigated the performance and origin of the single-phase perovskite oxides for hydrogen evolution reaction in alkaline media demonstrating the superior catalytic activity and the stability. The authors can identify the super-exchange interactions by Ru dopant in SrTiO₃ and the concentration of oxygen vacancies using experimental and theoretical calculation results. They showed the single-phase oxides can be more active than other reported composite materials possessing the different activate sites for reaction intermediates. The reviewer has following comments and questions that might bring discussion on the optimization and extensive understanding on their behavior.

Comments:

1. The authors confirmed that co-existence and modification of charge state on the surface of the catalyst can provide reaction sites for both OH⁻ adsorption and water dissociation. In this regard, the activity or efficiency could be influenced by the number or density of two different sites on the surface. It would be also interesting to analyze and demonstrate that Ru dopants are equivalently occupied in the bulk region and subsurface region of the catalyst from the experimental or DFT calculations.
2. The authors clearly showed that Ru doping on SrTiO₃ can be advantageous in presence of super-exchange behavior between Ru and Ti. The reviewer is also wondering if there are clue and hint predicting the similar behavior of other dopants to Ru ions with Ti ions. For example, can we expand the property of each component such as the orbital alignment of d-orbitals between Ti and Ru to HER catalytic property or adsorption behavior?

Reviewer #3 (Remarks to the Author):

In the manuscript, the authors have synthesized single-phase HER catalyst, SrTi_{0.7}Ru_{0.3}O_{3-δ} (STRO) perovskite oxide. Induced by a unique super-exchange interaction, the STRO perovskite exhibits outstanding HER activity with a low overpotential of 46 mV at 10 mA cm⁻² and Tafel slope of 40 mV dec⁻¹ in 1 M KOH. The super-exchange interaction in the STRO perovskite was proved and the underlying alkaline HER mechanism on STRO perovskite oxide was investigated via DFT calculations. However, some important concerns look strange or wrong, which could result from less rigorous experimental protocols. As such the paper cannot be published in Nature Communications. Some detailed questions are listed below:

1. The onset overpotential of Pt/C is lower than STRO, however, as the current density reaches about 40 mA/cm², the overpotential of Pt/C starts to be higher than STRO. Why would that happen? Moreover, the Tafel slope of Pt/C should be added to compare with other samples.
2. The shape of CV plots in the EDLC measurements (Figure S8a c) look strange/wrong. EDLC is

expected to observe a rectangular CV profile, in which the current should not change with the potential.

3. Since the Ti^{3+} and Ru^{5+} induced by super-exchange effect could promote the HER activity of STRO, then why with the amount of Ru increased to 0.4, the HER activity turned out to be inhibited?

4. As the authors demonstrated, the H_2O molecular is first absorbed on Ti^{3+} , but it is usually supposed that O is easier to be bonded with metal ions with higher valence. Why in this system O is first bonded with Ti^{3+} instead of Ru^{5+} ?

5. The language needs additional polishing.

Point-to-Point Responses to Reviewers' Comments and Suggestions

First of all, we thank the reviewers for their valuable comments and suggestions, which have resulted in modifications that have improved the quality and clarity of this paper. To address the specific concern/point clearly, we have separated the referees' comments into question areas and answered them in turn as follows.

Reviewer #1

GENERAL COMMENTS: This work presents a very active and novel electrocatalyst for the HER. The primary claim of novelty is that their STRO material acts as a single-phase synergistic electrocatalyst. The activity of this electrocatalyst is the highest of any perovskite electrocatalyst in the literature and amongst the highest non-precious metal electrocatalysts reported. The data supporting this claim are very strong and this catalyst does appear to be very exciting. I can see no issue with the experimental approaches used to collect this data and am impressed with the breadth of analysis used. I believe this paper contributes not only a very good electrocatalyst, but also a strong mechanistic and theoretical basis for this activity. The authors describe a new method through which one can optimise a perovskite electrocatalyst - that of synergistic coupling of two B-site cations and lattice oxygen / vacancies. For these reasons, I recommend publication of the manuscript in Nature Communications with some changes noted below.

Response:

We very much appreciate the Reviewer #1's highly positive comments and recommendation for publication. Our responses to his/her comments per point are as follows:

Comment 1: Page 4 (lines 85-88): The binding of OH⁻ should be optimal (neither too weak nor too strong), not necessarily as weak as possible as implied by the highlighted lines. (Dubouis, N. & Grimaud, A. The hydrogen evolution reaction: From material to interfacial descriptors. Chem. Sci. 10, 9165–9181 (2019))

Response to C1:

We thank the reviewer for pointing this out. We have modified related statements in the revised manuscript (**Line 87-88, Page 4**):

“The binding energy of OH should be optimal for a high-performance catalyst, so that more active sites can be exposed for the full reaction to proceed.”^{20b}

Comment 2: At the top of page 5 it is implied that this is a cost-effective catalyst. However, there is no cost comparison in the paper to Pt or other high-activity non-PGM electrocatalysts. Given that the reported catalyst has activity close to that of platinum, it is highly likely that including a cost-based metric would make this material even more impressive. (Zhu, Y. L. et al. Unusual synergistic effect in layered Ruddlesden - Popper oxide enables ultrafast hydrogen evolution. Nat. Commun. 10, 9 (2019)).

Response to C2:

We thank the reviewer for this valuable suggestion. As suggested by the reviewer, we have calculated the price activity of STRO, SRO, and commercial Pt/C catalysts for cost comparison as shown in **Fig. R1** (also **Fig. S12**). It can be seen that the STRO catalyst shows much higher price activity than the commercial Pt/C catalyst, highlighting its cost-effectiveness.

Fig. R1 (also **Fig. S12**). Mass activity (MA) and price activity (PA) of SRO, STRO, and Pt/C at the overpotential of $\eta = 0.1$ V.

Related data (**Fig. S12**) and discussions have been added in the revised manuscript as follows (please see the yellow-highlighted part in **Line 217-218, Page 11** and **Line 219-220, Page 12** in the revised manuscript):

“Furthermore, the mass activity (MA) and price activity (PA) of SRO, STRO, and Pt/C was also calculated (Supplementary Fig. 12). It can be seen that the STRO catalyst exhibits both much higher mass activity and price activity than the commercial Pt/C catalyst, demonstrating the cost-effectiveness of STRO catalyst in

practical applications.

Comment 3: The Tafel plot of the Pt/C electrocatalyst would be interesting as a comparison in Figure 3b. Generally, given that Pt/C is the current catalyst of choice, it would worth including it in D and E as well.

Response to C3:

We thank the reviewer for this valuable suggestion. As suggested by the reviewer, the Tafel plot of the commercial Pt/C has been added in the **Fig. 3b** in the revised manuscript. As shown in **Fig. R2**, the Pt/C catalyst possesses the Tafel slope of 40 mV dec⁻¹ in 1 M KOH, which is in accordance with previous studies (*Nat. Commun.*, 2019, 10, 631; *Energy Environ. Sci.*, 2019, 12, 149; *Energy Environ. Sci.*, 2019, 12, 2569; *Adv. Mater.*, 2018, 30, 1803676; *Adv. Energy Mater.*, 2018, 8, 1801690; *Adv. Funct. Mater.*, 2019, 29, 1901217). Regarding to the specific activity and TOF values of Pt/C, we would like to stress that the measurement of electrochemically active surface area (ECSA, required for specific activity and TOF) of metallic Pt is completely different from that of metal oxides based on previous reports (*Chem. Soc. Rev.*, 2019, 48, 2518; *Science*, 2009, 324, 1302; *Nat. Commun.*, 2019, 10, 149). The ECSA of metal oxides is estimated from the double-layer capacitance measurement, whereas the hydrogen underpotential deposition method is always applied for determining the ECSA of Pt materials. In view of the great difference between these two methods for calculating ECSA, so it is unfair to compare the specific activity and TOF between STRO oxide and Pt metal.

Fig. R2 (also **Fig. 3b**). The Tafel plots of STO, STRO, SRO and commercial Pt/C catalysts.

Comment 4: The arrow on Figure 3c is unnecessary and the trend it indicates appears to be rather weak.

Response to C4:

We thank the reviewer for pointing this out. As suggested by the reviewer, we have removed the arrow in the revised manuscript.

Comment 5: The figure caption for 3g states a ‘cathodic current density of -10 mA cm^{-2} ’. A cathodic current density by definition (at least according to IUPAC) is negative, so the minus sign is redundant.

Response to C5:

We thank the reviewer for pointing this out. As suggested by the reviewer, we have removed the minus in the revised manuscript.

Comment 6: Does the BET surface area give any useful activity? The BET measurement is made on the dry catalyst, whereas the electrocatalyst ink is a mixture of the perovskite, conductive carbon, Nafion. This would presumably greatly change the accessible surface area of the electrode. As such supplementary figure 9 which is normalized to the ECSA would perhaps be of more relevance than Figure 3d.

Response to C6:

We thank the reviewer for the valuable comments and suggestions. In fact, besides the ECSA, the specific activity normalized to the BET is also widely reported for metal oxides to reflect the intrinsic activity of catalysts (*Science*, 2011, 334, 1383; *J. Am. Chem. Soc.*, 2014, 136, 7077; *Adv. Mater.*, 2017, 1606800; *Angew. Chem. Int. Ed.*, 2016, 128, 5363; *Adv. Funct. Mater.*, 2017, 27, 1703455; *Nat. Commun.*, 2016, 7, 11510; *Nat. Commun.*, 2016, 7, 11053; *Angew. Chem. Int. Ed.*, 2015, 127, 3969; *Adv. Mater.*, 2016, 28, 6442.). As suggested by the reviewer, we have updated the **Fig. 3d** with specific activity normalized to the ECSA in the revised manuscript.

Comment 7: There are no mass-normalised current densities presented in the work. In

addition to current densities at given overpotentials, it is useful to provide mass activity ($A\text{ g}^{-1}$) to increase comparability with other materials.

Response to C7:

We appreciate for reviewer's valuable suggestions. As suggested by the reviewer, the mass-normalized activity comparison has been added in the revised manuscript as shown in **Fig. R1** (also **Fig. S12**). It can be seen that the STRO catalyst shows much higher mass activity than the SRO and Pt/C catalysts.

Related data (**Fig. S12**) and discussions have been added in the revised manuscript as follows (please see the yellow-highlighted part in **Line 217-218, Page 11** and **Line 219-220, Page 12** in the revised manuscript):

“Furthermore, the mass activity (MA) and price activity (PA) of SRO, STRO, and Pt/C was also calculated (Supplementary Fig. 12). It can be seen that the STRO catalyst exhibits both much higher mass activity and price activity than the commercial Pt/C catalyst, demonstrating the cost-effectiveness of STRO catalyst in practical applications.

Comment 8: The figure legend in supplementary figure 9 has labelled all the lines wrong.

Response to C8:

We thank the reviewer for pointing this out. As suggested by the reviewer, we have modified this error in the revised manuscript.

Comment 9: The downward arrow in Figure 6b doesn't add anything to the figure and makes reading the plot harder.

Response to C9:

We thank the reviewer for pointing this out. As suggested by the reviewer, we have removed the downward arrow in the revised manuscript.

Comment 10: The red and blue octahedra in Figure 6e should be addressed in the figure caption. Presumably the red octahedra refer to RuO_6 ?

Response to C10:

We thank the reviewer for this valuable suggestion. As suggested by the reviewer, the red and blue octahedra (RuO_6 and TiO_6) have been addressed in the figure caption in the revised manuscript.

Comment 11: In the experimental section you should state the size/area of the working electrode to aid reproducibility.

Response to C11:

We thank the reviewer for this valuable suggestion. As suggested by the reviewer, the size of working electrode (5 mm in diameter) has been provided in the revised manuscript.

Comment 12: A general note of the figures: The large half-filled circles used in many of the plots makes reading the data and seeing the error bars difficult. A small cross or similar would improve readability

Response to C12:

We thank the reviewer for this valuable suggestion. As suggested by the reviewer, we have changed the large half-filled circles into small crosses in the revised manuscript for improving readability.

Comment 13: Generally, this paper is well-written but there are several instances of poor/inaccurate grammar and wording which should be caught with a thorough proof-reading.

Response to C13:

We thank the reviewer for pointing this out. As suggested by the reviewer, we have carefully polished the language in the revised manuscript.

Reviewer #2

GENERAL COMMENTS: The authors investigated the performance and origin of the single-phase perovskite oxides for hydrogen evolution reaction in alkaline media demonstrating the superior catalytic activity and the stability. The authors can identify the

super-exchange interactions by Ru dopant in SrTiO₃ and the concentration of oxygen vacancies using experimental and theoretical calculation results. They showed the single-phase oxides can be more active than other reported composite materials possessing the different activate sites for reaction intermediates. The reviewer has following comments and questions that might bring discussion on the optimization and extensive understanding on their behavior.

Response:

We very much appreciate the Reviewer #2's valuable comments to improve our manuscript. According to the raised suggestions, more experiments and calculations have been made to clarify the issues and our point-by-point responses are as follows.

Comment 1: The authors confirmed that co-existence and modification of charge state on the surface of the catalyst can provide reaction sites for both OH⁻ adsorption and water dissociation. In this regard, the activity or efficiency could be influenced by the number or density of two different sites on the surface. It would be also interesting to analyze and demonstrate that Ru dopants are equivalently occupied in the bulk region and subsurface region of the catalyst from the experimental or DFT calculations.

Response to C1:

We thank the reviewer for the valuable comments and suggestions. In experiments, we firstly performed high-resolution HAADF-STEM to confirm the homogeneous distribution of Ru elements. As seen from the HAADF-STEM (**Fig. R3**), the Ru dopants homogeneously distribute in the surface and bulk regions. In addition, previous theoretical study suggested that Ru is thermodynamically stable on the surface and it can still be kinetically stabilized in sub-surface regions as well (*Energy Environ. Sci.* 2018, 11, 1762). As such, there will be a sizable and stable concentration of Ru centers within the topmost layers of the SrTiO₃ host to ensure a consistent and stable activity for STRO.

Fig. R3 (also **Fig. 2f**). HAADF-STEM and the corresponding elemental mapping images of STRO.

Comment 2: The authors clearly showed that Ru doping on SrTiO₃ can be advantageous in presence of super-exchange behavior between Ru and Ti. The reviewer is also wondering if there are clue and hint predicting the similar behavior of other dopants to Ru ions with Ti ions. For example, can we expand the property of each component such as the orbital alignment of d-orbitals between Ti and Ru to HER catalytic property or adsorption behavior?

Response to C2:

We thank the reviewer for the insightful comment and question. This is a truly valid point, as it allows for the generalization of the principles that dictate the presence of Ti³⁺ and M⁵⁺ on the surface of SrTiO₃. To check the possible universality, we have performed additional charge analysis for several other metal dopants on the surface of STO, e.g., Ir, Mo, Nb and Pt (**Fig. R4**). These metal ions tend to be high oxidation states in perovskite lattice. We found that the charges on Ti are 2.10, 2.04, 2.03 and 2.05 |e| for Ir, Mo, Nb and Pt dopants, respectively, which are very similar to the case (2.10 |e|) of Ru dopant, suggesting a charge transfer from the metal dopants to the surface Ti sites. This result could be a clear signal of the presence of Ti³⁺ along with M⁵⁺ surface species. While we have not carried out explicit calculations on the alkaline HER mechanism on these additional systems, which will constitute part of a broader experimental and computational screening work in the future.

Fig. R4 (also **Fig. S23**). Charge density redistribution upon the introduction of different metal dopants. From left to right: Ir, Mo, Nb and Pt.

To further experimentally confirm the positive role of $\text{Ti}^{3+}/\text{M}^{5+}$ exchange behavior in alkaline HER activity, we tried to prepare another Mo-doped $\text{SrTi}_{0.7}\text{Mo}_{0.3}\text{O}_{3-\delta}$ perovskite (denoted as STMO) and compared its activity with STO and SrMoO_4 (SMO). STMO exhibits higher catalytic activity than STO and SMO (**Fig. R5**), suggesting the super-exchange behavior between Mo and Ti (similar to STRO) can also boost alkaline HER. Based on above theoretical and experimental results, we believe that constructing $\text{Ti}^{3+}/\text{M}^{5+}$ couples in perovskites via super-exchange effect may be a universal way for boosting alkaline HER.

Fig. R5 (also **Fig. S25**). Polarization curves of STO, STMO, and SMO catalysts in an Ar-saturated 1 M KOH solution. Scan rate, 5 mV s^{-1} .

Related data (**Fig. S23-25**) and discussions have been added in the revised manuscript as follows (please see the yellow-highlighted part in **Line 368-382, Page 20**):

“Inspired by the key role of super-exchange behavior between Ru and Ti in STRO system in promoting alkaline HER activity, we were intrigued to find out whether other dopants into STO can also possess similar phenomenon. First, we performed additional charge analysis for several other metal dopants on the surface of STO, e.g., Ir, Mo, Nb and Pt (Supplementary Fig. 23). It was found that the charges on Ti are 2.10, 2.04, 2.03 and 2.05 |e| for Ir, Mo, Nb and Pt dopants, respectively, which are very similar to the case (2.10 |e|) of Ru dopant, suggesting a charge transfer from the metal dopants to the surface Ti sites. This result could be a clear signal of the presence of Ti^{3+} along with M^{5+} surface species. To further experimentally confirm the positive role of Ti^{3+}/M^{5+} exchange behavior in alkaline HER activity, we tried to prepare one Mo-doped $SrTi_{0.7}Mo_{0.3}O_{3-\delta}$ perovskite (denoted as STMO) and compared its activity with STO and $SrMoO_4$ (SMO) (Supplementary Fig. 24 & 25). STMO exhibits higher catalytic activity than STO and SMO, suggesting the super-exchange behavior between Mo and Ti (similar to STRO) can also boost alkaline HER. Based on above theoretical and experimental results, we believe that constructing Ti^{3+}/M^{5+} couples in perovskites via super-exchange effect may be a universal way for boosting alkaline HER.”

Reviewer #3

GENERAL COMMENTS: In the manuscript, the authors have synthesized single-phase HER catalyst, $SrTi_{0.7}Ru_{0.3}O_{3-\delta}$ (STRO) perovskite oxide. Induced by a unique super-exchange interaction, the STRO perovskite exhibits outstanding HER activity with a low overpotential of 46 mV at 10 mA cm^{-2} and Tafel slope of 40 mV dec^{-1} in 1 M KOH. The super-exchange interaction in the STRO perovskite was proved and the underlying alkaline HER mechanism on STRO perovskite oxide was investigated via DFT calculations. However, some important concerns look strange or wrong, which could result from less rigorous experimental protocols. As such the paper cannot be published in Nature Communications. Some detailed questions are listed below:

Response:

We sincerely appreciate the constructive comments raised by the Reviewer#1 to improve our manuscript. Following the suggestions, we have made more experiments and explanations to clarify the issues and our point-by-point responses are as follows.

Comment 1: The onset overpotential of Pt/C is lower than STRO, however, as the current density reaches about 40 mA/cm², the overpotential of Pt/C starts to be higher than STRO. Why would that happen? Moreover, the Tafel slope of Pt/C should be added to compare with other samples.

Response to C1:

We thank the reviewer for raising this question. We also thank the reviewer for drawing our attention to this phenomenon that “The onset overpotential of Pt/C is lower than STRO, however, as the current density reaches about 40 mA/cm², the overpotential of Pt/C starts to be higher than STRO.” In fact, such phenomenon was widely reported in previous studies (*Chem*, 2018, 4, 1139; *Nat. Commun.*, 2018, 9, 2452; *Adv. Mater.*, 2014, 26, 2683; *Adv. Funct. Mater.*, 2020, 30, 2000551; *J. Mater. Chem. A*, 2020, 8, 10831; *Angew. Chem. Int. Ed.*, 2020, 59, 1659; *J. Mater. Chem. A*, 2020, 8, 9239; *Angew. Chem. Int. Ed.*, 2016, 128, 703; *J. Mater. Chem. A*, 2020, 8, 17202; *Small*, 2017, 13, 1701648), which can be explained by ***the underwater superaerophobic surface of STRO*** (*Chem*, 2018, 4, 1139; *Nat. Commun.*, 2018, 9, 2452; *Adv. Mater.*, 2014, 26, 2683; *J. Mater. Chem. A*, 2020, 8, 10831). Experimental result shows that the contact angle of the gas bubble on STRO in 1 M KOH solution is $151.2 \pm 2.0^\circ$, which is much larger than that ($91.5 \pm 3.0^\circ$) on the Pt/C electrode (**Fig. R6**). Such a superaerophobic surface of STRO can significantly weaken the bubble effect that is especially prominent at large current densities, promote the mass transfer process during HER (i.e., facilitate the quick leaving of as-formed hydrogen bubbles and the electrolyte diffusion), and thereby result in a better catalytic activity at large current densities than Pt/C. Because of the fast kinetics of the HER on platinum, the current density is usually limited by the mass-transport of H₂ generated under high overpotentials. (*Chem. Sci.*, 2019, 10, 9165; *J. Phys. Chem. B*, 1997, 101, 5405; *J. Electrochem. Soc.*, 2015, 162, F190). This may explain why the activity of STRO dramatically exceeds that of Pt/C at high current densities. Besides, as suggested by the reviewer, the Tafel plot of the commercial Pt/C has been added in **Fig. 3b** in the revised manuscript.

Fig. R6 (also **Fig. S8**). The contact angle of hydrogen bubbles with **a.** STRO and **b.** Pt/C in 1 M KOH solution.

Related data (**Fig. S8**) and discussions have been added in the revised manuscript as follows (please see the yellow-highlighted part in **Line 173-177, Page 9**):

*“Noticeably, the HER current of STRO can largely exceed that of the benchmark Pt/C catalyst beyond -0.08 V, which can be ascribed to the underwater superaerophobic surface of STRO”. The superaerophobic surface of STRO as reflected by high contact angle of the gas bubble (**Supplementary Fig. 8**) can promote the quick leaving of as-generated H₂ bubbles and facilitate mass-transport especially at large current densities^{32,33}.*

Comment 2: The shape of CV plots in the EDLC measurements (Figure S8a c) look strange/wrong. EDLC is expected to observe a rectangular CV profile, in which the current should not change with the potential.

Response to C2:

We thank the reviewer for this comment. We agree with the reviewer that EDLC in an ideal system is expected to observe a rectangular CV profile. EDLC measurements are always carried out within the non-Faradaic potential regime. In our study, the current slightly change with the potential in the EDLC measurements, but the shape of CV plots still look close to rectangular shape. *In fact, such phenomenon that the current change with the potential in the CV profiles for EDLC measurements was widely reported in many previous studies (Energy Environ. Sci., 2019, 12, 2569; Angew. Chem. Int. Ed., 2019, 58, 11796; J. Phys. Chem. C, 2016, 120, 27746; Nat. Commun., 2019, 10, 149; Adv. Energy Mater., 2017, 7, 1601390;*

Adv. Energy Mater., 2018, 8, 1801690; *Angew. Chem. Int. Ed.*, 2019, 58, 5432; *Joule*, 2017, 1, 383; *Energy Environ. Sci.*, 2019, 12, 572; *ACS Catal.*, 2018, 8, 5431; *ACS Nano*, 2016, 10, 8851; *Angew. Chem. Int. Ed.*, 2019, 58, 461; *Adv. Mater.*, 2018, 30, 1707105; *ACS Energy Lett.*, 2018, 3, 1360; *Adv. Energy Mater.*, 2018, 8, 1703538; *ACS Catal.*, 2017, 7, 7131; *Nat. Commun.*, 2020, 11, 2720, etc.), and ***catalysts with low electrochemical surface area always displays such phenomenon***. The detailed reasons for such phenomenon are currently not very clear, which may be associated with the material properties and electrochemistry; future in-depth investigation could shed more light on this and it is beyond the scope of this work.

Comment 3: Since the Ti^{3+} and Ru^{5+} induced by super-exchange effect could promote the HER activity of STRO, then why with the amount of Ru increased to 0.4, the HER activity turned out to be inhibited?

Response to C3:

We thank the reviewer for raising this question. As demonstrated in our work, the STRO catalyst is endowed with ***multiple catalytic sites for affecting overall alkaline HER activity***, including Ti^{3+} , Ru^{5+} , and oxygen vacancy, which is not only associated with Ru^{5+} . In other words, the overall alkaline HER activity of STRO is dominated by multiple active sites. With the amount of Ru increased to 0.4, it means the amount of active sites Ti^{3+} and oxygen vacancies will be reduced, which would have negative influence on both the water dissociation and H^* adsorption steps during HER. Therefore, the HER activity turned out to be inhibited for STR0.4O. The optimal HER activity of STR(0.3)O among STRO system could be due to the perfect combination in the amount of multiple catalytic sites (e.g., Ti^{3+} , Ru^{5+} , and oxygen vacancy/lattice-oxygen).

Comment 4: As the authors demonstrated, the H_2O molecular is first absorbed on Ti^{3+} , but it is usually supposed that O is easier to be bonded with metal ions with higher valence. Why in this system O is first bonded with Ti^{3+} instead of Ru^{5+} ?

Response to C4:

We thank the reviewer for raising this question. The notion that negative O ions are generally easier to be bonded with metal ions with higher valence holds true ***when the metal ions only***

separately exist. However, in perovskite oxide system which contains metal ions coordinated with O^{2-} and oxygen vacancy, this notion is not necessarily correct. For perovskite oxides, the oxygen binding is not only related to the valence of metal ions, but also associated with the *coordination environment of metal ions*. We take $SrRuO_3$ and $SrTiO_3$ as examples for illustration. Although the formal oxidation state of both TMs is +4, Ti and Ru exhibit different oxygen binding strengths. A slightly different example is, if one creates an oxygen vacancy which leads to the reduction of the two neighboring sites in perovskite, then we normally find that the binding energy of O (H_2O) is stronger on metal site with lower valence (i.e., defected surface) than on the higher oxidation site (i.e, surface without defects) (*J. Phys. Chem. C*, 2017, 121, 8378; *Surf. Sci.*, 2015, 633, 38; *J. Mater. Chem.*, 2011, 21, 18983).

Specifically in our case, *the enhanced O (H_2O) binding of Ti^{3+} than Ru^{5+} in STRO perovskite may stem from the way the TM d-orbitals are split and occupied*. The metal environments are shown in **Fig. R7**, which depict the octahedral arrangement with a point group symmetry of O_h with the usual t_{2g}/e_g splitting of d-orbitals as well as the surface TM sites where the point group symmetry is C_{4v} . *For Ti^{3+} which is a $3d^1$ system, the electron occupies a d_{yz} orbital which favors interaction with H_2O* , whereas, the Ru^{5+} which is a $4d^3$ system has its frontier states occupying a d_{xy} orbital which does not provide an optimal overlap the p orbitals of an incoming O or H_2O molecule (*J. Phys. Chem.*, 1983, 87, 2960; *Angew. Chem. Int. Ed.*, 1987, 26, 846).

Fig. R7. Bulk and surface coordinates of the transition metals showing the splitting and occupancies of the d-orbitals.

Comment 5: The language needs additional polishing.

Response to C5:

We thank the reviewer for pointing this out. As suggested by the reviewer, we have carefully polished the language in the revised manuscript.

REVIEWER COMMENTS

Reviewer #1 (Remarks to the Author):

The authors have addressed the comments well and I recommend publication. I only note that the legend in Supplementary Fig. 12 should read PA (not SA) for the green bars.

Reviewer #2 (Remarks to the Author):

The authors have carefully reviewed and responded the questions from reviewers by conducting additional computations and experiments. I believe that the manuscript is now revised supports their findings clearly and strengthening arguments.

Reviewer #3 (Remarks to the Author):

Dai et al reported a highly efficient STRO electrocatalysts for HER in alkali conditions. This catalyst assembles the active sites of water dissociation, OH desorption and H* adsorption in a single compound. Regarding the efficient HER electrocatalysts in alkali conditions are generally composed of hybrid systems, this represents certain progress. However, the abuse of "super exchange" should be avoided.

Some comments:

1. The conception of "super-exchange" may be re-checked and explained it in the introduction part.
2. In page 15 line 266, the "the generation of the Ti³⁺ and Ru⁵⁺ ions produced from the super exchange" seems to be incorrect. It should be other way around. Super exchange is the effect rather than the cause.
3. The super exchange is generally observed in antiferromagnetic system. The direct evidence by magnetic susceptibility measurement is not provided.
4. In Page 16 line 292, "the super-exchange induced oxygen vacancy generation" is rather confusing.
5. The reduction induced enhancement of HER activity of R-STO may not be simply due to the formation of oxygen vacancy. In this case, the "super exchange" may be indeed enhanced.

Point-to-Point Responses to Reviewers' Comments and Suggestions

First of all, we thank the reviewers for their valuable comments and suggestions, which have resulted in modifications that have improved the quality and clarity of this paper. To address the specific concern/point clearly, we have separated the referees' comments into question areas and answered them in turn as follows.

Reviewer #1

GENERAL COMMENTS: The authors have addressed the comments well and I recommend publication. I only note that the legend in Supplementary Fig. 12 should read PA (not SA) for the green bars.

Response:

We very much appreciate the Reviewer #1's recommendation for publication. Regarding one minor revision, as suggested by the reviewer, the legend of SA in Supplementary Fig. 12 has been corrected to be PA.

Reviewer #2

GENERAL COMMENTS: The authors have carefully reviewed and responded the questions from reviewers by conducting additional computations and experiments. I believe that the manuscript is now revised supports their findings clearly and strengthening arguments.

Response:

We very much appreciate the Reviewer #2's recommendation for publication.

Reviewer #3

GENERAL COMMENTS: Dai et al reported a highly efficient STRO electrocatalysts for HER in alkali conditions. This catalyst assembles the active sites of water dissociation, OH desorption and H* adsorption in a single compound. Regarding the efficient HER electrocatalysts in alkali conditions are generally composed of hybrid systems, this represents certain progress. However, the abuse of "super exchange" should be avoided:

Response:

We sincerely appreciate the Reviewer #3's highly positive evaluation with "this represents certain progress" on our work. Regarding inappropriate use "super exchange" in some places, we apologize for our negligence in these places and appreciate the reviewer's constructive comments to improve our manuscript. Following the suggestions, we have made more explanations and revisions to clarify the issues and our point-by-point responses are as follows.

Comment 1: The conception of "super-exchange" may be re-checked and explained it in the introduction part.

Response to C1:

We thank the reviewer for this valuable suggestion. As suggested by the reviewer, the conception of "super-exchange" in this work have been re-checked and explained in the introduction part. Related discussions have been added in the revised manuscript as follows (please see the yellow-highlighted part in **Line 109-113, Page 5**):

"Remarkably, an unusual super-exchange effect in STRO perovskite was discovered, which contributes to the 180° interaction between neighboring Ti^{3+} (with a $3d^1$ configuration) and Ru^{5+} (with a $4d^3$ configuration) ions according to the Goodenough-Karamori-Anderson rule, and enhanced electrical conductivity."

Comment 2: In page 15 line 266, the "the generation of the Ti^{3+} and Ru^{5+} ions produced from the super exchange" seems to be incorrect. It should be other way around. Super exchange is the effect rather than the cause.

Response to C2:

We thank the reviewer for this valuable comment. We agree with the reviewer that the super-exchange is the effect rather than the cause, and we have modified this inappropriate phrase in the revised manuscript to be "*the generation of the Ti^{3+} and Ru^{5+} ions via the super-exchange effect*". (**Line 267, Page 15**)

Comment 3: The super exchange is generally observed in antiferromagnetic system. The direct evidence by magnetic susceptibility measurement is not provided.

Response to C3:

We thank the reviewer for pointing this out. Actually, the super-exchange interactions of *d* electrons in a pair of transition metals can be well understood by **Goodenough-Karamori-Anderson (GKA) rule** (*J. Phys. Chem. Solids*, 1959, 10, 87-98). As seen from the Table 2 and Table 5 in the reference of *J. Phys. Chem. Solids*, antiferromagnetic (AFM) or ferromagnetic (FM) super-exchange depends on **the number of d electrons and M-O-M band angle**. According to the GKA rule, the 180° interaction between Ti^{3+} (with a $3d^1$ configuration) and Ru^{5+} (with a $4d^3$ configuration) in STRO is expected to be FM, which was also reported in a similar $SCo_{1-x}Ru_xO_{3-\delta}$ system (*J. Phys. Condens. Matter*, 2016, 28, 436005; *Small*, 15, 2019, 1903120).

Table 2. The 180° interaction between cations in octahedral sites

Number of 3d-electrons of interacting cations	Species of interacting cations	Relevant bond and mechanism	Resultant superexchange interaction	Total super-exchange interaction
d^3-d^3	$Mn^{4+}-Mn^{4+}$ $Cr^{3+}-Cr^{3+}$	σ -bond and π -bond A, G, A-H, S	Antiferro.	Antiferro.
d^8-d^8	$Ni^{2+}-Ni^{2+}$	σ -bond A, G, A-H, S	Antiferro.	Antiferro.
d^5-d^5	$Mn^{2+}-Mn^{2+}$ $Fe^{3+}-Fe^{3+}$	σ -bond A, G, A-H, S π -bond G, A-H, S π -bond A	Antiferro. Antiferro. (weak) Uncertain (weak)	Antiferro.
d^8-d^3	$Ni^{2+}-V^{2+}$	σ -bond and π -bond A, G, A-H, S	Ferro.	Ferro.
d^5-d^3	$Fe^{3+}-Cr^{3+}$	σ -bond A, G, A-H, S π -bond G, A-H π -bond A, S	Ferro. Antiferro. (weak) Uncertain (weak)	Ferro.
d^4-d^4	$Mn^{3+}-Mn^{3+}$	*		
d^6-d^8	FeO	σ -bond A, G, A-H, S π -bond	Antiferro. Uncertain (weak)	Antiferro.
d^7-d^7	CoO	σ -bond A, G, A-H, S π -bond	Antiferro. †	Antiferro.

* Depends on the direction of the line of superexchange.

† Weak, but dependent on the direction of the line of superexchange.

A=ANDERSON'S mechanism, G=GOODENOUGH'S mechanism, A-H=ANDERSON and HASEGAWA'S mechanism, S=SLATER'S mechanism.

Cited from the reference (*J. Phys. Chem. Solids*, 1959, 10, 87-98)

Table 5. The 90° interaction between cations in octahedral sites

Number of 3d-electrons of interacting cations	Species of interacting cations	Relevant bond and mechanism	Resultant superexchange interaction	Total super-exchange interaction
d^8-d^8	$Ni^{2+}-Ni^{2+}$	$p\sigma-dy$ bond A, G S $s-dy$ bond A, G, A-H, S	Ferro. Uncertain Antiferro.	Ferro.
d^5-d^5	$Mn^{2+}-Mn^{2+}$ $Fe^{3+}-Fe^{3+}$		Uncertain	*
d^3-d^3	$Cr^{3+}-Cr^{3+}$	$p\sigma-dy$ and $p\sigma-d'e'$ A, G, A-H, S $p\pi-d'e'$ and $s-dy'$ A, G, A-H, S	Ferro. Antiferro. (weak)	Ferro.
d^8-d^8	$Ni^{2+}-V^{2+}$	$p\sigma-dy$ and $p\sigma-d'e'$ A, G, A-H S $p\pi-d'e'$ and $s-dy', s-dy'$ A, G, A-H	Antiferro. Uncertain (weak) Ferro. (weak)	Antiferro.

* Tendency towards antiferromagnetic interaction with decreasing number of 3d-electrons.

Cited from the reference (*J. Phys. Chem. Solids*, 1959, 10, 87-98)

Comment 4: In Page 16 line 292, “the super-exchange induced oxygen vacancy generation” is rather confusing.

Response to C4:

We thank the reviewer for this valuable comment. As suggested by the reviewer, we have deleted the phrase of “super-exchange induced” and modified the sentence to be “*the oxygen vacancy generation and electrical conductivity enhancement in STRO are demonstrated*”.

(Line 292-293, Page 16)

Comment 5: The reduction induced enhancement of HER activity of R-STO may not be simply due to the formation of oxygen vacancy. In this case, the “super exchange” may be indeed enhanced.

Response to C5:

We thank the reviewer for this valuable comment. We agree with the reviewer that the reduction induced enhancement of HER activity of R-STO may not be simply due to the formation of oxygen vacancy, which may be also associated with the enhanced super-exchange interaction. As suggested by the reviewer, we have added related sentence in the revised manuscript: “*Besides, the enhanced super-exchange interaction in R-STO may also contribute to increased activity.*” (yellow-highlighted in Line 347-348, Page 19)

REVIEWERS' COMMENTS

Reviewer #3 (Remarks to the Author):

This manuscript have been carefully revised by the authors. All the comments have been addressed. I believe that the manuscript is now suitable for publication in Nature Communications.

Point-to-Point Responses to referee's comments and suggestions

First of all, we thank you for your valuable comments and suggestions, which have resulted in modifications that have improved the quality and clarity of the paper. To address the specific point clearly, we have separated the referees' comments into question areas and answered them in turn as follows:

Reviewer #3:

This manuscript has been carefully revised by the authors. All the comments have been addressed. I believe that the manuscript is now suitable for publication in Nature Communications.

Response: We thank the reviewer for the positive assessment and recommendation for publication in Nature Communications.